# Utilization of Microwaves: A Novel Approach to SARS-CoV-2 Genome Extraction

**Marta Elena Álvarez-Argüelles** (ID)**, Susana Rojo-Alba, Gabriel Martín** * (ID)**, Zulema Pérez-Martínez, Cristian Castelló-Abietar, Jose Antonio Boga and Santiago Melón** (ID)

Hospital Universitario Central de Asturias, Av. Roma, s/n, 33011 Oviedo, Spain
* Correspondence: gabrielmartinrguez1994@gmail.com

**Abstract:** In this study, an innovative approach to the heat extraction method has been tested: the use of microwaves, which can dramatically decrease the time that is needed to do the genome extraction. The method can obtain the virus with enough quality to assure the identification by RT-qPCR and minimize procedures and contaminations.

**Keywords:** SARS-CoV-2; genome extraction; in-house method; heat extraction; microwaves

## 1. Introduction

With the global SARS-CoV-2 pandemic, the need for quick and efficient techniques to process samples has become an aim for all the clinical laboratories around the world. Several methods such as heat extraction can be a clear alternative to improve the time that is taken for the procedure at the time of doing the genome extraction [1–4].

In this study, an innovative approach to the heat extraction has been tested: the use of microwaves, which can dramatically decrease the time that is needed to do the genome extraction. The method can obtain the virus with enough quality to assure the identification by q(RT)-PCR and minimize procedures and contaminations.

## 2. Material and Methods

To check this hypothesis, a set of 70 nasopharyngeal samples (1 mL), 45 of them obtained from patients that were previously confirmed as SARS-CoV-2 positive by de Microbiology Service of the Hospital Universitario Central de Asturias (being the rest negative), were extracted by five different methods: two of them using chaotropic reagents, and three based on heat extraction. Prior to each extraction procedure, the samples were correctly homogenized using a IKA Vortex3 (Sigma-Aldrich, St. Louis, MO, USA) for 30 s.

These samples were extracted in 5 batches of 14 samples (9 positives and 5 negatives in random positions) plus one positive control and one negative control for all the procedures.

The two methods that use reagents were automatic MagNa Pure 96 (Roche, Ginebra, Switzerland) taken as a reference, and the "Bikop" method that was developed in our laboratory previously [3]. Both procedures were carried with 200 μL of each sample, obtaining 100 μL of eluted product.

About the three methods that were based on heat extraction, one consisted just of a heat application at 98 °C for 10 min as the normal method. The second method was based on the addition of 25 μL of proteinase K (pK) to favor denaturalization before heat application; for this method, the heat application had two steps: first at 56 °C during 5 min followed by other at 98 °C for 10 min [5–7]. These two methods were performed in a SureCycler 8800 (Agilent Technologies, Santa Clara, CA, USA).

The last method of heat extraction was the use of microwaves. This method was performed on a microwave oven Schneider SMW205 (Schneider Electric, Llanera, Spain), during 1 min at 800 W and 2450 MHz of microwave frequency. For all these protocols, the

heat application was followed by a cooling step on ice for 5 min. A volume of 100 μL of each sample was used for each of the heat extraction protocols.

All of the extracted samples were tested with a multiple RT-qPCR that was directed to two regions of the SARS-CoV 2 genome (Orf1ab and N gene). Briefly, 5 μL of sample, that was previously extracted by any of the tested methods, were added to 10 μL of TaqMan Fast 1-Step Master Mix (Life technologies, Carlsbad, CA, USA) that was supplemented with a mixture of primers (Thermo Fisher Scientific, Walthman, MA, USA) and TaqMan MGB probes (Applied Biosystems, Foster City, CA, USA) which were the same conditions as used in reference [3]. The amplification and subsequent analysis were carried out using the Applied Biosystems 7500 Real-time PCR System (Applied Biosystems). The cycling protocol was as follows: (50 °C, 20 min; 95 °C, 5 min; 45 cycles of 95 °C, 10 s; and 55 °C, 15 s and 60 °C, 30 s).

### 3. Results

These data (Figure 1) show that three protocols that were tested had over 95% sensitivity (100% for MP96 and 95.34% for the others). The microwaves method reached an 88.37% sensitivity. No negative sample was found positive for SARS-CoV-2. There were two of the positive samples that were tested that were found non-valuable.

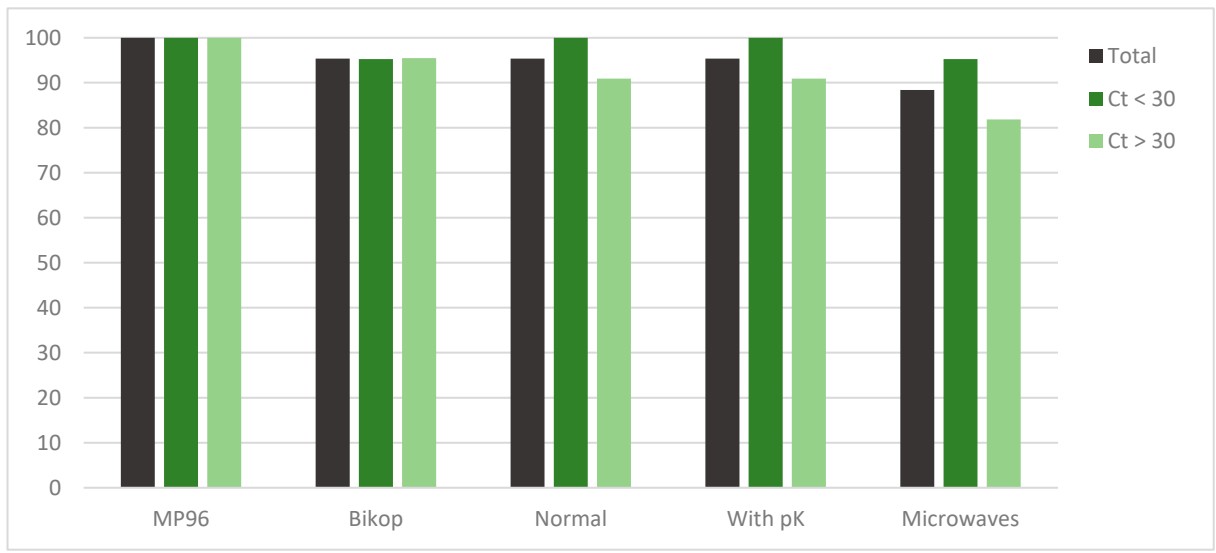

**Figure 1.** Representation of the total sensitivity of each method and distributed by cycle threshold subgroups.

After a T-Student analysis (α = 0.05), comparing each method with the reference method, significant *p*-values were obtained for the microwave method, specifically for total samples (0.04) and for samples over Ct 30 (0.02), meaning that these sensitivities were statistically different from those that were obtained on the reference method MP96.

The cycle threshold for each sample of each procedure and the statistical results, such as the sensitivity, of the total and broken down into less than and greater or equal to Ct 30 are shown in Tables 1 and 2.

**Table 1.** Sensitivity, mean, range, IC95%, and *p*-value for the different methods that were tested.

| | Extraction with Reagents | | Heat Extraction | | |
|---|---|---|---|---|---|
| | **MP96** | **Bikop** | **Normal** | **With pK** | **Microwaves** |
| Total | | | | | |
| Positives | 43 (100%) | 41 (95.34%) | 41 (95.34%) | 41 (95.34%) | 38 (88.37%) |
| $\overline{X} \pm \sigma$ | 29.07 ± 4.44 | 27.29 ± 4.36 | 26.39 ± 4.57 | 25.58 ± 4.02 | 27.31 ± 5.02 |
| Range | (21–34) | (15–37) | (17–36) | (18–35) | (17–37) |
| IC95% | (27.86–30.28) | (25.99–28.59) | (25.03–27.75) | (24.38–26.78) | (25.81–28.81) |
| *p*-value | - | 0.20 | 0.21 | 0.21 | 0.04 |
| Ct < 30 | | | | | |
| Positives | 21 (100%) | 20 (95.24%) | 21 (100%) | 21 (100%) | 20 (95.24%) |
| $\overline{X} \pm \sigma$ | 25.57 ± 2.68 | 25.15 ± 4.30 | 24.14 ± 4.22 | 24.85 ± 4.14 | 26.3 ± 5.39 |
| Range | (21–29) | (15–32) | (17–31) | (18–30) | (17–35) |
| IC95% | (24.43–26.71) | (23.31–26.99) | (22.33–25.95) | (23.08–26.62) | (23.99–28.61) |
| *p*-value | - | 0.23 | 0.45 | 0.49 | 0.45 |
| Ct ≥ 30 | | | | | |
| Positives | 22 (100%) | 21 (95.45%) | 20 (90.91%) | 20 (90.91%) | 18 (81.82%) |
| $\overline{X} \pm \sigma$ | 32.4 ± 1.37 | 29.33 ± 3.40 | 28.75 ± 3.70 | 26.35 ± 3.86 | 28.44 ± 4.45 |
| Range | (30–34) | (23–37) | (22–36) | (21–35) | (20–37) |
| IC95% | (31.83–32.97) | (27.91–30.75) | (27.20–30.30) | (24.74–27.96) | (26.58–30.30) |
| *p*-value | - | 0.19 | 0.09 | 0.10 | 0.02 |

**Table 2.** Cycle threshold for each sample by genome extraction procedure.

| Sample | Extraction with Reagents | | Heat Extraction | | |
|---|---|---|---|---|---|
| | **MP96** | **Bikop** | **Normal** | **Adding pK** | **Microwave** |
| 1 | 21 | 27 | 21 | 22 | 19 |
| 2 | 21 | 20 | 17 | 18 | 17 |
| 3 | 23 | 15 | 18 | 21 | 21 |
| 4 | 23 | 21 | 24 | 30 | 29 |
| 5 | 23 | 28 | 26 | 30 | 26 |
| 6 | 23 | 19 | 18 | 18 | 18 |
| 7 | 24 | 25 | 23 | 27 | 32 |
| 8 | 24 | 27 | 30 | 30 | 27 |
| 9 | 25 | 25 | 23 | 27 | 31 |
| 10 | 25 | 27 | 23 | 29 | 29 |
| 11 | 25 | 29 | 30 | 24 | 24 |
| 12 | 26 | 25 | 23 | 29 | 29 |
| 13 | 27 | 25 | 21 | 21 | 35 |
| 14 | 27 | 30 | 28 | 26 | 32 |
| 15 | 28 | 31 | 29 | 24 | 0 |
| 16 | 28 | 25 | 29 | 23 | 34 |
| 17 | 28 | 22 | 21 | 21 | 22 |
| 18 | 29 | 22 | 21 | 21 | 24 |
| 19 | 29 | 0 | 25 | 30 | 26 |
| 20 | 29 | 28 | 26 | 22 | 21 |
| 21 | 29 | 32 | 31 | 29 | 30 |
| 22 | 30 | 26 | 28 | 21 | 24 |
| 23 | 30 | 23 | 23 | 22 | 26 |
| 24 | 30 | 28 | 26 | 24 | 25 |
| 25 | 31 | 28 | 0 | 29 | 31 |
| 26 | 31 | 32 | 29 | 27 | 26 |
| 27 | 31 | 31 | 32 | 35 | 30 |
| 28 | 32 | 27 | 31 | 27 | 35 |
| 29 | 32 | 29 | 31 | 32 | 37 |
| 30 | 32 | 26 | 22 | 23 | 25 |

**Table 2.** *Cont.*

| Sample | Extraction with Reagents | | Heat Extraction | | |
|:---:|:---:|:---:|:---:|:---:|:---:|
| | **MP96** | **Bikop** | **Normal** | **Adding pK** | **Microwave** |
| 31 | 33 | 31 | 0 | 28 | 0 |
| 32 | 33 | 32 | 25 | 22 | 20 |
| 33 | 33 | 0 | 35 | 28 | 27 |
| 34 | 33 | 27 | 26 | 23 | 31 |
| 35 | 33 | 25 | 30 | 22 | 28 |
| 36 | 33 | 30 | 29 | 0 | 28 |
| 37 | 33 | 27 | 26 | 32 | 26 |
| 38 | 33 | 31 | 32 | 26 | 26 |
| 39 | 34 | 29 | 36 | 0 | 0 |
| 40 | 34 | 31 | 28 | 25 | 31 |
| 41 | 34 | 37 | 28 | 26 | 36 |
| 42 | 34 | 36 | 32 | 30 | 0 |
| 43 | 34 | 30 | 26 | 25 | 0 |
| 44 | NV | NV | NV | NV | NV |
| 45 | NV | NV | NV | NV | NV |
| 46 | 0 | 0 | 0 | 0 | 0 |
| 47 | 0 | 0 | 0 | 0 | 0 |
| 48 | 0 | 0 | 0 | 0 | 0 |
| 49 | 0 | 0 | 0 | 0 | 0 |
| 50 | 0 | 0 | 0 | 0 | 0 |
| 51 | 0 | 0 | 0 | 0 | 0 |
| 52 | 0 | 0 | 0 | 0 | 0 |
| 53 | 0 | 0 | 0 | 0 | 0 |
| 54 | 0 | 0 | 0 | 0 | 0 |
| 55 | 0 | 0 | 0 | 0 | 0 |
| 56 | 0 | 0 | 0 | 0 | 0 |
| 57 | 0 | 0 | 0 | 0 | 0 |
| 58 | 0 | 0 | 0 | 0 | 0 |
| 59 | 0 | 0 | 0 | 0 | 0 |
| 60 | 0 | 0 | 0 | 0 | 0 |
| 61 | 0 | 0 | 0 | 0 | 0 |
| 62 | 0 | 0 | 0 | 0 | 0 |
| 63 | 0 | 0 | 0 | 0 | 0 |
| 64 | 0 | 0 | 0 | 0 | 0 |
| 65 | 0 | 0 | 0 | 0 | 0 |
| 66 | 0 | 0 | 0 | 0 | 0 |
| 67 | 0 | 0 | 0 | 0 | 0 |
| 68 | 0 | 0 | 0 | 0 | 0 |
| 69 | 0 | 0 | 0 | 0 | 0 |
| 70 | 0 | 0 | 0 | 0 | 0 |

NV = No Valuable.

## 4. Discussion

Viral load, reflected as Ct, has an influence on these results. When a threshold was made in Ct, all the procedures that were analyzed were sensitive enough for samples which were amplified under 30 cycles, with results over 95% for every method. Over 30 Ct, or even more, over 32 Ct, a decrease in sensitivity happens, especially with the microwaves method.

As it has been published before by different authors, Ct values have an indirect association with viral load, so a high Ct value means a low viral load [8,9]. Knowing this, as the data show that the undetected positive samples correspond to high cycles (so low viral loads) that are not related to viral transmissibility [10], we can accept that, even when the microwave method sensitivity is 88.37%%, it is high enough to assure it use for SARS-CoV-2 detection.

About the statistically significant difference that was obtained for the total sensibility of the microwaves method, it is surely due to the lower sensitivity that was obtained on the Ct over 30 subgroup, so this group weighs the sensitivity of the total.

On the other hand, it is noted that manual methods have higher variance than the automatized reference methods, as the ranges of each sample are higher than in the MP96 method. Special care must be taken when manual methods are performed.

In conclusion, the results confirm that heat extraction can be implemented in a clinical laboratory as a fast method to get samples ready for q(RT)-PCR and offers earlier detection. The use of the microwaves method can be the fastest way of extraction when the number of samples exceeds the capacity of the laboratory to process them in time, especially when high incidence is noted, or in cases of a shortage of extraction reagents.

**Author Contributions:** Conceptualization, M.E.Á.-A.; Methodology, S.R.-A. and G.M.; Validation, S.R.-A.; Formal analysis, Z.P.-M.; Investigation, C.C.-A.; Data curation, G.M.; Writing-original draft preparation, G.M.; Writing-review and editing, J.A.B. and S.M. All authors have read and agreed to the published version of the manuscript.

**Funding:** This research received no external funding.

**Institutional Review Board Statement:** This study was approved by Comité de Ética de la Investigación del Principado de Asturias with code CEImPA 2021.188.

**Data Availability Statement:** Not applicable.

**Acknowledgments:** Thanks to ASCOL (Asturiana de Control Lechero) for their financial support for this research.

**Conflicts of Interest:** The authors declare no conflict of interest.

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
