# Peer review of "Utilization of Microwaves: A Novel Approach to SARS-CoV-2 Genome Extraction"

_2673-8007, doi:10.3390/applmicrobiol2040060_

Round 1

Reviewer 1 Report (Previous Reviewer 1)

The study entitled "Utilisation of microwaves: a novel approach to SARS-CoV-2 genome extraction" is important research. The authors have incorporated the previous suggestions. However, I found still various points where manuscript needs to revise and updated very carefully. 

Is there any way if the results can be presented in the form of figures?

I believe, giving more information regarding the selection of the subjects would be beneficial. 

Line no. 11: change to In this study

Line no. 46: and that the microwaves method almost get 90%., reframe this sentence and write it separately for the clarity.

Line no. 54: change to  as it has been

Line no. 56 to 58 , the information is unclear, reframe the sentences to increase the readability.

Likewise, English need to be improved at various places. 

Best wishes

Author Response

The authors appreciate all de comments of the reviewer. Now, we proceed to answer them point by point:

  1. First, a graphic with the sensitivities of each method (total and by cycle threshold ranges) has been added to the manuscript.

  1. We must precise that all the information we have is that these samples were from patients with COVID-19 suspicion All the samples were anonymized before being processed by the Microbiology Service of our hospital, and we work with them for this study after the normal work routine and validation of results.

  1. Spelling mistakes on line 11 has been corrected.

  1. Line 46 has been reformulated for a better comprehension.

  1. Spelling mistakes on line 54 has been corrected.

  1. All the paragraph has been reformulated for a better comprehension.

  1. English spelling mistakes has been reviewed and corrected.

Reviewer 2 Report (Previous Reviewer 2)

Alvarez-Arguelles et al tested a new method that uses microwaves to extract RNA from 70 nasopharyngeal samples.
45 out of 70 samples were confirmed as positive for SARS-COV-2 by the microbiology service of the Central University Hospital of Asturias.
This new approach should be a faster method than other methods tested and minimize the time for procedures in diagnostic practices.

In the revised manuscript, the authors improved the description of sample handling by making the description of the method of preparing samples for analysis clearer.

However, some points should be considered before the publication:

In line 47 the authors state that the differences are not significant, but in Table 1 the p-values ​​for the microwave protocol were 0.04 and 0.02 for the total samples and for the samples with Ct> 30, respectively.
In general, the differences are considered statistically significant with p-values ​​almost less than 0.05. This point should be explained.
How were the p-values ​​reported in Tables 1 calculated? The methods used for the statistical analysis must be indicated.

Author Response

The authors appreciate all de comments of the reviewer. For sure the reviewer has reason when appointing the lack of information about the statistical treatment of our data and the confusion on the brief mention of them.

We have added now how p-values were calculated and an explanation for the microwaves p-values obtained mentioned by the reviewer.

Reviewer 3 Report (New Reviewer)

The manuscript entitled “ Utilisation of microwaves: a novel approach to SARS-CoV-2 genome extraction” 

Studied the heat extraction of genomes from SARS-Cov-2 samples. The author utilized five different methods: two of them utilized the reagents, the rest of them were based on the heat extraction. 

The author should clarify some major concerns:

1.The complete manuscript should include the abstract, introduction, results, and discussion. But the authors didn’t provide the abstract, the merged section of introduction, results, and the discussion. 

2. The authors didn’t describe the heat methods in detail, which is one of the most important sections for this manuscript. I.e. Line 19, the author didn’t mention the homogenization, including the brand of the instrument and the parameters for setting up the vortex.  

3. There are many grammatical or spelling errors. For example,

Line 1: “Utilisation of microwaves…” should be “Utilization of ….” The author should carefully proofread the manuscript. 

Author Response

The authors appreciate all de comments of the reviewer. Now, we proceed to answer them point by point:

  1. The manuscript was written at first as a brief report, not as a common research article, that the cause it has not differentiated parts, but, as the reviewer suggest, we have established the mentioned sections.

  1. Authors want to clarify that heat extraction methods were reference on the manuscript with the bibliography, but for a clearer comprehension of the methods, we have added the thermocycler were normal and with pK heat extractions methods were performed, the volume of pK added to samples and the vortex used for homogenization.

  1. English spelling mistakes has been reviewed and corrected.

Round 2

Reviewer 3 Report (New Reviewer)

The authors addressed the concerns.

This manuscript is a resubmission of an earlier submission. The following is a list of the peer review reports and author responses from that submission.

Round 1

Reviewer 1 Report

The present study entitled "Microwaves: a novel approach to SARS-CoV-2 genome extraction" is an interesting piece of the research. However,  I have some suggestions which can be considered before the publication.

1. The title must be "Utilisation of the microwaves: A novel approach to SARS-CoV-2 genome extraction " this will give a more clear picture to the readers. 

2. Change SARS-CoV2 to SARS-CoV-2

3. In the report it has been mentioned:

To check this hypothesis, a set of 43 nasopharyngeal samples were extracted by five different methods: two of them using chaotropic reagents, and three based on heat extraction. The two methods that use reagents were automatic MagNa Pure 96 (Roche, Ginebra) taken as reference, and the “Bikop” method developed on our laboratory previously (3). Both procedures were carried with 200µL of each sample.

My question is from which subjects/people the samples have been collected

Are these patients detected with COVID-19?

4. In the article authors mentioned that the: Viral load, reflected as Ct values

I suggest to discuss more about this association. How viral load can be interpreted from the Ct values.

The following article can be utilised 

Rabaan AA, Tirupathi R, Sule AA, Aldali J, Mutair AA, Alhumaid S, Muzaheed, Gupta N, Koritala T, Adhikari R, Bilal M, Dhawan M, Tiwari R, Mitra S, Emran TB, Dhama K. Viral Dynamics and Real-Time RT-PCR Ct Values Correlation with Disease Severity in COVID-19. Diagnostics. 2021; 11(6):1091. https://doi.org/10.3390/diagnostics11061091

5. Special care must be token when manual methods

Check the spelling in the above line, change it to taken  

6. In the conclusion authors mentioned use of microwaves method can be the fastest way

It is better to justify this with some data and figures in the conclusion

or just provide a highlight of your data which can justify the significance of your work. 

Author Response

Authors wish to thank all the comments expressed by the reviewer. Now, we answer it one by one:

  1. Change on title has been considered and accepted.
  2. Spelling mistake has been corrected.
  3. About question 3, yes, all the patients were confirmed positive to SARS-CoV-2 by the Microbiology Service before they were used for this assay. This information has been added to the manuscript.
  4. Considerations on question 4 about relation ship between cycle threshold and viral load has been added to the manuscript, surely enriching it.
  5. Spelling mistake has been corrected.
  6. A better expression of the value of our results, that highlights the benefits of the microwave method has been added to the manuscript, hopping it will satisfy the reviewer interest.

Reviewer 2 Report

Alvarez-Arguelles et al tested a new method that uses microwaves to extract RNA from nasopharyngeal samples containing the SARS-COV-2 virus. This new approach should be a faster method than other methods tested and minimize the time for procedures in diagnostic practices.
Some points of the RNA extraction procedure and the results obtained are unclear.
The authors tested 43 nasopharyngeal samples for viral RNA extraction using 5 different methods / kits. It is unclear how they used the same sample with 5 different methods. Did they split each sample into 43 parts? Are the authors sure that they have homogeneous samples of each sample containing the same amount of virus? How do they verify that this condition is met? The authors should explain in detail how they handle the samples for use with 5 different extraction methods.
The conclusions are contradictory to the reported data. The authors indicate that 1) the four methods are "over 95% of sensitivity, except for the microwaves method, that almost get 90%."; and 2) The microwave method has a positive rate of 38 out of 43 samples, not detecting 5 positive samples. This means 5 false negatives for the microwave method, with a higher false negative rate than other methods. These two assertions are in contrast with the conclusions that indicate this new extraction method useful for preventing "the elongation of the contagious chain."
In my opinion, to counter the spread of the COVID-19 outbreak it is more important that the diagnostic procedure identifies all the positives rather than being faster without identifying all the subjects infected with SARS-COV-2.
Furthrmore, The study lacks negative samples to evaluate microwave specificity and 4 other methods. Negative samples should be added to the study to determine the specificity of the microwave methods compared to 4 other methods.

minor point:
The authors use a qPCR protocol listed in the reference as number 3, which is a pre-print manuscript. To give greater strength to the quantification procedures with qPCR, authors should refer to a method reported in a manuscript published in a peer-reviewed journal.

Author Response

Authors appreciate all the comments of the Reviewer.

For the used of each sample, which has a total volume of 1mL, we have used directly 200µL for the MagNa Pure 96 and the Bikop methods (obtaining 100µL of elution product) and 100µL for the 3 heat extraction methods. Previous to the extraction, samples were correctly homogenized using a vortex during 30 seconds. This clarification has been added to the manuscript for better comprehension.

About the second point, authors totally understand the point of view of the reviewer, but want to clarify that, all the positive samples not detected by the microwaves method are samples with high Ct values, related to low viral loads and less transmissibility of the virus by patients (this information has been added to the manuscript), so, even when its true that microwaves method is not perfect detecting all positives, the ones that are not detectable are the less significant.

For the last comment, It is true that the use of negative samples is not mentioned on the manuscript, a mistake that authors regret. A set of 25 negative samples were used during the study, this data was not considered relevant at first, but after the reviewer comment we have note our mistake and added this information to the manuscript.

About the minor point, the references 3 listed as pre-print was recently accepted and published by Journal of Virological Methods and has been actualized.